# Improvement of Heat-Treated Wood Coating Performance Using Atmospheric Plasma Treatment and Design of Experiments Method

**DOI:** 10.3390/polym13091520

**Published:** 2021-05-09

**Authors:** Ender Hazir

**Affiliations:** Department of Forest Industry Machine and Management, Faculty of Forestry, Istanbul University-Cerrahpaşa, Bahcekoy, Istanbul 34473, Turkey; ender.hazir@iuc.edu.tr

**Keywords:** multiobjective optimisation, colour change, adhesion strength, desirability function, factorial design, artificial weathering test

## Abstract

The aim of this work is to improve the heat-treated wood coating performance using experimental design methodology and air–plasma treatment. Firstly, two different heat treatment processes were applied to the wood samples. In the second stage of the study, air–atmospheric plasma treatment was applied to heat-treated samples. These samples were coated with water-based varnish. Adhesion strength and colour change values of these samples before and after the artificial weathering test were measured. The design of experiments method was used to investigate the significant factors. The heat treatment process (212 °C—1 h and 212 °C—2 h) and atmospheric plasma treatment parameters (pressure, distance, and feed) were selected as independent variables, while adhesion strength and colour change were determined as dependent variables. The factors affecting the surface coating performance before and after the artificial weathering test were evaluated by analysis of variance (ANOVA) and Pareto plot. In addition, the factor levels that maximise the adhesion strength value and minimise the colour change were found using the multiobjective optimisation technique. According to the multiobjective optimisation method, results of treatment feed, working distance, and pressure of 60 mm/s, 7.69 mm, and 1 bar were considered as optimum plasma treatment conditions, respectively, for heat treatment process A. Corresponding values for the heat treatment process B were 60 mm/s, 10 mm, and 2 bar.

## 1. Introduction

Water-based paint applications are widely used in wooden materials due to their environmentally friendly properties [1,2]. In particular, it offers significant advantages in increasing the resistance of heat-treated wood materials against external weather conditions, providing colour stability and creating an aesthetic appearance. The main purpose of the surface coating is to protect the appearance the properties of wood materials and to prevent physical and chemical deterioration due to weathering conditions [3,4]. With the application of surface coating, it is possible to increase the service life of wood materials. Although many tests such as layer hardness, scratch, abrasion, and gloss are used to determine the performance of coating materials, colour change and adhesion strength tests are among the most crucial tests [5,6,7,8].

In recent years, there have several techniques to increase the durability of the wood material. The most widely applied methods are heat treatment, impregnation, acetylation, and thermally modification [9,10]. The heat treatment process causes chemical changes in cell wall polymers, decreases hydroxyl groups, increases hydrophobicity, and causes a decrease in water absorption. Since no chemical process can be applied in this technique, it is an environmentally friendly method, providing dimensional stability, weathering durability, and increasing resistance to biological effects. For this reason, heat-treated wood materials offer significant advantages, especially for outdoor use, and have a wide range of uses [11,12,13,14]. In addition to these advantages, some negative effects occur due to the exposure of the wood material to different temperatures and durations. The negative changes in the mechanical properties of the wood material and the adhesion problems of water-based paints or adhesives to the surface depending on the wettability [15,16,17]. At the same time, colour changes occur depending on the adhesion problem of the coating to the surface. These performance tests, namely, colour change and adhesion strength, were a crucial indicator of the durability of the coating and the quality of the surface coating, especially in outdoor applications [18]. The colour change that occurs, especially in exterior applications, is considered an important quality problem. Although there are many factors affecting the colour change, the least colour change is desired in applications.

Improving the adhesion resistance and discolouration by applying environmentally friendly water-based varnish to heat-treated wood materials will increase the service life of such materials. Especially, radio frequency, corona discharge, and atmospheric plasma methods, which have been used since the early 1970s, continue to work on improving the surface characteristics of wood material. Plasma treatment is a common modification technique that uses ionised gas to change the surface properties of materials. The plasma state, known as the fourth state of matter, can be defined as a partially ionised gaseous medium consisting of electrons, ions, photons, and various neutral species at many different excitation levels that can physically or chemically interact with organic matter [19,20,21]. Especially, increasing the wettability of wood material is among the priority activities [22,23].

It is known that this application changes the polymer properties of the material such as the wettability and adhesion of the material. The main areas of application are adhesives, printing, and extrusion coating. In addition, good results are expected in transferring to wood [22,24,25,26,27,28,29].

Moreover, when the wood material is exposed outside without any protection, it causes complex physicochemical changes in the wood surface [30,31]. These changes mainly start with solar radiation and the leaching of degradation products. Other factors such as moisture, heat, abrasion caused by particles blown by the wind, atmospheric pollution, oxygen, and human activities are also effective in wood degradation [32].

Modification of wood and wood-based materials with air plasma treatment can reduce or eliminate these negative properties of wood material. This process is widely used to improve the wettability, fluid uptake, or adhesion properties of wood [33,34,35]. In several studies, it has been concluded that plasma treatment mainly increases the polar component of the surface energy and positively affects the wood surface properties [36,37,38,39,40]. The main components in wood are lignin, hemicellulose, and cellulose, which make up about 95% of the wood mass. Klarhofer et al. showed the oxidation of lignin (formation of oxygenated functional groups) and a reduction in cellulose (loss of hydroxyl groups, formation of C/O double bonds) as a result of plasma treatment [41]. Wolkenhauer improved the properties of wood/polyethylene and wood/polypropylene composites modified with atmospheric pressure and ambient air plasma to increase adhesion properties [42].

Highly hydrophobic maple surfaces were prepared by atmospheric dielectric barrier discharge (DBD) processes with ethylene, methane, chlorotrifluoroethylene, and hexafluoropropylene precursors [43]. The hydrophobicity of these plasma-based studies increased despite the fact that they were fully modified wood surfaces, with all reported static water contact angles (WCA) below 145 the generally accepted 150 thresholds for superhydrophobicity [44,45,46].

According to literature studies show that plasma application factors such as treatment speed, working distance, pressure, application time, power, and frequency were effective in changing the characteristic properties of the material surface. In addition to these, the properties vary with the geometry and frequency of the electrode induced by plasma and corona discharge [47,48,49,50,51,52,53].

For this reason, it is necessary to systematically investigate the change caused by the plasma application parameters on the material. One of the most widely used methods in this research is an experimental design methodology. This method is used in research and optimisation of the effects of many independent changes on the dependent variable. In particular, factorial designs have widespread use, both in investigating the interactions of independent variables with each other and in reducing the cost of the experiment [54,55,56].

In this study, the 2^k^ experimental design method was used to investigate the effects of independent variables on dependent variables. For the optimisation of independent variables, the desirability function-based multiobjective optimisation method was used.

## 2. Materials and Methods

### 2.1. Properties of Wood Material

Sapele wood (*Entandrophragma cylindricum*) materials were supplied by Akdeniz Forest Products Company (Istanbul, Turkey), and samples were provided according to the principles specified in ASTM D-358 [57]. Special attention was paid to choose the wood material supplied from logs without knots, no buckling, and no growth defects. These specimens were chosen due to having a wide range of uses in exterior applications. Dimensions of the specimens were determined as 21 mm (radial) × 120 mm (tangential) × 1000 mm (longitudinal). These specimens were conditioned in climate at (20 ± 2) °C and (65 ± 1)% relative humidity (RH) until they reach an equilibrium moisture content. After the samples reached equilibrium moisture, the heat treatment process was applied in a controlled manner. Heat chamber factors were determined as temperature (212 °C) and duration (1 h and 2 h). In this case, heat treatment process parameters were selected (212 °C—1 h) and (212 °C—2 h).

### 2.2. Air–Plasma Treatment Process

After completing the heat treatment process on wood materials at different temperatures and times, air–plasma modification was applied to these samples at different parameter levels. Atmospheric plasma is a method used to improve the characteristics of the material surface. It is used to improve the performance of the coating applied to the surface by cleaning and activating. The most important equipment of the plasma system is the high voltage power supply and the type of nozzle. The properties of the plasma system (open air) used in the study are given in Table 1.

Air–plasma treatment process transfers the electrodes to the plasma by passing the compressed air. Plasma treatment process factors and levels are given in Table 2.

### 2.3. Wood Coating Process

After completing the heat and plasma treatment process on wood materials, water-based varnish, materials of which were supplied by AkzoNobel (Istanbul, Turkey), was applied to the samples. The varnish application process was prepared according to ASTM-D 3023 (2017) principles and application instructions [58]. Varnish processing was carried out in three stages, namely, primer, filling, and topcoat. At all the stages of the varnishing process, the sanding dust formed was removed from the wooden surfaces with an air gun. The application process was carried out from a distance of 20–30 cm with a spray gun with a bottom chamber nozzle of 1.8–2.2 mm and air pressure of 2–2.5 bar. At the primer application, the prepared samples were sanded with 80 and 150 numbered sandpaper. After this process was completed, approximately 150 g/m^2^ of varnish was applied to the samples. At the filling and topcoat varnish application, after these samples were kept at 20 °C and 65% relative humidity for 4 h, the samples were intermediate sanded with a grit size of 220. Approximately, 130 g/m^2^ filling varnish and 150 g/m^2^ topcoat varnish were applied to the samples. The varnish properties used in the primer, filling, and topcoat applications are given in Table 3.

### 2.4. The 2^k^ Factorial Design

The experimental design methodology is widely used in many engineering problems. This method is especially used to investigate the relationship between dependent and independent variables and to determine the optimum factor levels. At the same time, this method, while reducing the experimental cost, allows the experiment to be carried out more systematically and in a shorter time. In this study, the 2^k^ experimental design method was chosen. This method can reveal the interactions between factors while investigating the independent variables that affect the output in a process (Minitab Statistical Software 17, Minitab Ltd., PA, USA).

The coded equation of 2^k^ factorial design is given in Equation (1) as follows:(1)$$y=\beta_{0}+\sum_{i=1}^{k}\beta_{i}\chi_{i}+\sum_{1\leq i\leq j}^{k}\beta_{ij}\chi_{i}\chi_{j}+\varepsilon$$
where the terms of $\beta_{i}$, $\beta_{0}$, $\beta_{ij}$, $\chi_{i}$ and $\chi_{j}$ were symbolised regression coefficient, average response, random error, the interaction between $\chi_{i}$ and $\chi_{j}$, independent variables [59].

In this study, heat treatment process such as process A (212 °C—1 h) and process B (212 °C—2 h), plasma treatment parameters such as treatment speed, working distance (Distance between the plasma head and wood surface), and pressure were selected as independent variables; adhesion strength and colour change were determined as dependent variables. Process factors values were gathered by using the 2^k^ factorial design. The coded variables as (−1) and (+1) are given in Table 4.

### 2.5. Determination of Coating Performance

#### 2.5.1. Determination of Adhesion Strength

One of the most significant quality characteristics showing the surface coating performance of wood and wood-based materials is the bonding of the coating to the surface, in other words, the adhesion strength of the coating material. This value is a crucial indicator of the durability of the coating and the quality of the surface coating, especially in exterior applications. In this study, the pull-off test method was used to measure the adhesion strength (PosiTest-AT, DeFelsko, New York, NY, USA). This test was carried out according to the EN ISO 4624-2016 standard [60]. In order to measure the adhesion strength, 20 mm diameter dollies were used with two-component silane–epoxy resin, and this application was made with 20 °C and 40% RH ambient conditions. After waiting 7 days, the adhesion strength of the samples was measured by using the test device.

#### 2.5.2. Determination of Colour Change

One of the most significant quality characteristics that show the surface coating performance of wood and wood-based materials is the colour change of the coating. In this study, the colour change of coated surfaces was determined using dual-beam spectrophotometer (D65 light source and an observer angle of 10°). CIE–Lab colour space coordinate system with L (lightness), a (red–green axis), and b (yellow–blue axis) was used to determine the colour values. The total colour change was calculated using Equation (2a–d):(2a)$$\Delta E^{*}=\sqrt{{\left(\Delta L^{*}\right)}^{2}\;+\;{\left(\Delta a^{*}\right)}^{2}+{\left(\Delta b^{*}\right)}^{2}}$$
Δ*L** = *L** after weathering − *L** before weathering(2b)
Δ*a** = *a** after weathering − *a** before weathering(2c)
Δ*b** = *b** after weathering − *b** before weathering(2d)

#### 2.5.3. Artificial Weathering Test

An artificial weathering test was used to perform the colour change and adhesion strength performances of heat-treated samples and plasma-treated coated wood samples. Briefly, 48 samples were prepared with dimensions of 140 × 75 × 5 mm^3^. These samples were exposed to artificial weathering tests according to the principles of TS EN ISO 16474-3 standard (Atlas UV 2000, Atlas Material Testing Technology, IL, USA) [61]. This application was carried out in the UV tester (8-340 UV-A lamps, BPT 40–110 °C, 104–230 °F, BST 40–120 °C, 104–248 °F, Single deck). Test application and control periods were determined as one cycle (6 h), interim check (500 h), and test duration (727 h).

## 3. Results

The results of adhesion strength and colour change before and after the artificial weathering test are given in Appendix (see Appendix A, Table A1).

### 3.1. Results of Adhesion Strength before the Artificial Weathering Test

#### 3.1.1. Results of ANOVA

The 24 factorial design and ANOVA were employed to determine the main effect and two-three-way interaction effects. Values of *F* and *p*-values of “prob > *F*” are lower than 0.05 showing that the equation terms are significant. The variables A, B, C, AC, BC, and ACD were effective factors on the adhesion strength. The model performance parameters were found as 95.55% (*R*-square) and 93.47% (*Adj-R*-square). The result of ANOVA for adhesion strength is given in Table 5.

#### 3.1.2. Evaluation of Pareto Plot

Figure 1 shows the Pareto chart of the standardised effects on the adhesion strength. The vertical line in the Pareto chart displays the statistically significant effect on the adhesion strength for a 5% significance level [59]. Any effect that ranges past this datum point is potentially important. The Pareto plot verified that the main effects of A, B, C and the interactions of AC, BC, and ACD were statistically significant at the 5% level on the adhesion strength.

#### 3.1.3. Evaluation of Main Effect Plot

The main effects parameters are displayed in Figure 2. Since the slope of variables such as process, feed, and pressure was steeper, the adhesion strength value was affected by each level of factors. A lower treatment feed, lower pressure, and process (A) were resulted with maximum adhesion strength. These results were confirmed by ANOVA analysis.

#### 3.1.4. Evaluation of Interaction Effects

The two-way interactions plot is displayed in Figure 3; this plot explains the one factor with an impact on other factors. The interaction effect between process and pressure indicates that they were significant parameters for the adhesion strength. Adhesion strength increased with lower pressure and process (A). The interaction effect between treatment feed and pressure indicates that they were significant parameters for the adhesion strength. Adhesion strength increased with lower pressure and lower treatment feed.

### 3.2. Results of Adhesion Strength after the Artificial Weathering Test

#### 3.2.1. Results of ANOVA

Table 6 shows the analysis of variance for adhesion strength. Values of “prob > F” are lower than 0.05, showing that the equation terms are significant. The variables A, B, C, AC, AB, ACD, and ABC were effective factors on the adhesion strength. The model performance parameters were found as 94.14% (*R*-square) and 91.40% (*Adj-R*-square).

#### 3.2.2. Evaluation of Pareto Plot

Figure 4 displayed the Pareto chart of the standardised effects on the adhesion strength. The Pareto plot verified that the main effects of A, B, C, and the interactions of AC, AB, ACD, and ABC were statistically significant at the 5% level on the adhesion strength.

#### 3.2.3. Evaluation of Main Effects

The main effects variables are displayed in Figure 5. Since the slope of variables such as process type, treatment feed, and pressure was steeper, the adhesion strength value was affected by each level of factors. A lower treatment feed, lower pressure, and process type (B) provided the maximum adhesion strength. These results were confirmed with ANOVA analysis.

#### 3.2.4. Evaluation of Interaction Effects

From Figure 6, the interaction effect between treatment feed and process type indicates that they were significant parameters for the adhesion strength. Adhesion strength increased with lower treatment feed and process type (A). The interaction effect between process type and pressure indicates that the two were significant parameters for the adhesion strength. Adhesion strength increased with lower pressure and process type (B). Adhesion strength increased with lower treatment time and lower temperature.

### 3.3. Results of Colour Change

#### 3.3.1. Results of ANOVA

The 24 factorial design and ANOVA were employed to determine the main effect and two-three-way interaction effects. Values of “prob > *F*” are lower than 0.05 showing that the equation terms are significant. The variables A, B, C, BC, and ABC were effective factors on the adhesion strength. The model performance parameters were found as 79.94% (*R*-square) and 70.54% (*Adj-R*-square). The results of ANOVA for colour change are given in Table 7

#### 3.3.2. Evaluation of Pareto Plot

Figure 7 displays the Pareto chart of the standardised effects on the colour change. According to the Pareto chart, A, B, C and the interactions of BC and ABC were statistically significant factors at the 5% level on the colour change.

#### 3.3.3. Evaluation of Main Effects

The main effects variables are displayed in Figure 8. Since the slope of variables such as process type, treatment feed, pressure, and working distance was steeper, the colour change value was affected by each level of factors. A lower treatment feed, lower pressure, lower distance, and process type (B) provided the minimum colour change. These results were confirmed by ANOVA analysis.

#### 3.3.4. Evaluation of Interaction Effects

From Figure 9, the interaction effect between pressure and treatment feed rate indicates that they were significant parameters for the colour change. Colour change decreased with lower treatment feed when the pressure was lower. The interaction effect between pressure process type and treatment feed also indicates that they were significant parameters for the colour change.

### 3.4. Validation of the Model

To test the normality assumption, it was determined by drawing the histogram of the residuals and the normal probability diagram. As seen in Figure 10, Figure 11 and Figure 12, there is no reason to suspect any violation of the independent or constant assumption of variance since the errors are distributed along a straight line and the errors appear to be a normal distribution. The residuals *d_i_* were computed with Equation (3):(3)$$d_{i}=\frac{e_{ij}}{\sqrt{MS_{E}}}$$
where $\sqrt{MS_{E}}$ and *e_ij_* are symbolised the mean error sum of squares and residuals.

### 3.5. Multiobjective Optimisation of Process Parameters

As stated in literature studies, there are many tests in determining the performance of surface coating wood materials. However, adhesion strength and colour change are among the most significant tests used to evaluate the surface coating performance of wood. Therefore, in this study, different from the literature studies, the factor levels that provide maximum adhesion strength and minimum colour change were optimised. The optimum values of the factor levels were investigated using the multiobjective optimisation technique. In this study, an artificial weathering method was used to compare surface coating performance values. Since this test is important in demonstrating performance results, colour change and adhesion strength data after artificial weathering test were used.

The desirability function is commonly used to search optimum variables levels for engineering problems. It has various values between 0 and 1. The aim of this function is to bring the desirability value closer to 1. In this work, the transformation of adhesion strength value was selected as the higher-the-better quality characteristic (see Equation (4a)), while the transformation of colour change value was selected as the lower-the-better quality characteristic (see Equation (4b)). These values are computed by using Equation (4a,b):(4a)$$\begin{array}{l} d_{i}=\left\{ \begin{array}{l} \;1\;\;\;y_{i}<T\\ {\left(\frac{U-y_{i}}{U-T}\right)}^{w}\;T\leq y_{i}\leq U\\ \;0\;\;\;y_{i}>U \end{array}\right.\; \end{array}$$
(4b)$$\begin{array}{l} d_{i}=\left\{ \begin{array}{l} \;0\;\;\;y_{i}<L\\ {\left(\frac{y_{i}-L}{T-L}\right)}^{w}\;L\leq y_{i}\leq T\\ \;1\;\;\;y_{i}>T \end{array}\right.\; \end{array}$$
where *T*, *L*, and *W* symbolise the target value of the *i*-th output, *y_i_* qualifies the acceptable lower limit value, and W symbolises the weight.

Maximise Adhesion strength (*AS*): (feed (*f*), pressure (*p*), distance (*d*));

Minimise Colour change (*CC*): (feed (*f*), pressure (*p*), distance (*d*));

Process A:max *AS* (9.46 − 0.0522 *f* – 2.476 *p* − 0.223 *d* + 0.02296 *fp* + 0.00252 *fd* + 0.138 *pd* − 0.00149 *fpd*)(5a)
min *CC* (−12.77 + 0.2534 *f* + 14.73 *p* − 0.208 *d* − 0.1794 *fp* + 0.0030 *fd* + 0.016 *pd* + 0.00012 *fpd*)(5b)

Process B:max *AS* (1.202 + 0.01075 *f* + 1.079 *p* − 0.0260 *d −* 0.01619 *fp* + 0.000531 *fd* − 0.0069 *pd* − 0.000031 *fpd*)(6a)
min *CC* (15.50 − 0.149 *f −* 6.46 *p* − 0.26 *d* + 0.1088 *fp* + 0.0056 *fd* + 0.075 *pd* − 0.00217 *fpd*)(6b)

Presented functions were optimised within the specified range applied in Table 4 and were Equation (7a–c).
60 ≤ *f* ≤ 100(7a)
1 ≤ *p* ≤ 2(7b)
4 ≤ *d* ≤ 10(7c)

In Figure 13, the minimum colour change and the maximum adhesion strength values for process A resulted in 1.69 MPa and 6.43, respectively. In Figure 14, the minimum colour change and the maximum adhesion strength values for process B resulted in 1.94 MPa and 5.32, respectively. The desirability values (*d*) for process A and process B were found as 0.8401 and 0.7742, respectively.

By virtue of the examination in the experimental design, working distance, treatment feed, and pressure values of 7.69 mm, 60 mm/s, and 1.00 bar were found as optimum air–plasma treatment parameters for process A. Corresponding values for process B were 10 mm, 60 mm/s, and 2.00 bar.

## 4. Discussion

The results obtained in Section 3.1 and Section 3.2 were compared with literature studies. In literature studies, it has been stated that there were chemical-, visual-, and surface-related changes in the material as a result of exposure of the wood material coated or noncoated surface treatment under outdoor weather conditions [60,61,62,63]. In the present study, coated wood samples were exposed to an artificial weathering test. Colour change was observed in the samples. In the present study, the water-based varnish was applied to the samples obtained by heat treatment and plasma modification. In determining the surface coating performance of these samples, colour change and adhesion resistance test results were evaluated. It has also been found in the literature that the plasma modification process changes the adhesion strength and the parameters of the plasma treatment process have an effect on the adhesion strength, and this is similar to the results of the present study. [48,50,51,53,64,65]. In addition, it has been found in the literature that the conditions of heat treatment are also effective on the adhesion strength and this is similar to the results of the present study. The colour change results obtained in Section 3.3 were compared with literature studies. In previous studies, it was concluded that colour change was affected by many factors and that the heat treatment process and plasma modification applications were effective on colour change. In particular, the result that the colour change in plasma applications was reduced [51]. In the present study, it was found that the colour change was reduced with the change of plasma application parameter levels.

In addition, it has been concluded that the interaction between the heat treatment conditions and the application of plasma modification parameter levels has an effect on the colour change. According to the results, when the adhesion strength performance before the artificial weathering test was evaluated, it was found that the heat treatment process type, treatment feed, and working distance were significant factors. It has also been shown that interactions between factors AC, BC, and ACD are effective variables. As a result of the Pareto plot, the adhesion strength was mostly affected by heat treatment process type. When the result of the main effect plot was evaluated, the higher adhesion strength was found at lower treatment feed, lower pressure, lower working distance, and process type A. When the adhesion strength performance after the artificial weathering test was evaluated, it was found that the heat treatment process type, treatment feed, and working distance were significant factors. It has also been shown that interactions between factors AC, AB, and ACD are effective variables. As a result of the Pareto plot, the adhesion strength was the most affected by ABC interaction. When the result of the main effect plot was evaluated, the higher adhesion strength was found at lower treatment feed, lower pressure, lower working distance, and process type B. According to the colour change results, the colour change was directly affected by the heat treatment process type and pressure. At the same time, interactions between factors ABC and AC were found to be effective on colour change. When the main effect plot was evaluated, the lower colour change was found in the heat treatment process type B, lower treatment speed, lower pressure, and lower working distance. Both before and after the artificial weathering test, when the models used for the adhesion strength results were statistically evaluated, R-square and Adj-R square values had high results. This result means that the independent variables explain the dependent variables sufficiently. However, when the colour change was evaluated statistically, even if the dependent variables of the independent variables were observed at a sufficient level, this value can be increased by adding different independent variables to the model. This situation shows that the colour change also has a complex structure.

## 5. Conclusions

The 2^4^ factorial design and multiobjective optimisation method were used to find the significant variables and optimum factor levels. Four process parameters such as treatment feed, working distance, and pressure were selected as continuous variables with the heat-treated process type selected as discrete variables.

When the results were evaluated, the parameters of the heat treatment process applied were found to be significant factors. Therefore, optimum factor levels of each process were calculated.

Results of treatment feed, working distance, and pressure of 60 mm/s, 7.69 mm, and 1 bar were considered as optimum plasma treatment conditions, respectively, for heat treatment process A. Corresponding values for the heat treatment process B were 60 mm/s, 10 mm, and 2 bar. In this case, the maximum adhesion strength and minimum colour change for process A were found as 6.43 MPa and 1.69, respectively. The maximum adhesion strength and minimum colour change for process B resulted in 5.32 MPa and 1.94, respectively. In addition, it was calculated that there was a 17.26% decrease in adhesion strength with increasing the treatment time under constant temperature. At the same time, it was found that there is an increase of 14.79% in the colour change.

## Figures and Tables

**Figure 1 polymers-13-01520-f001:**
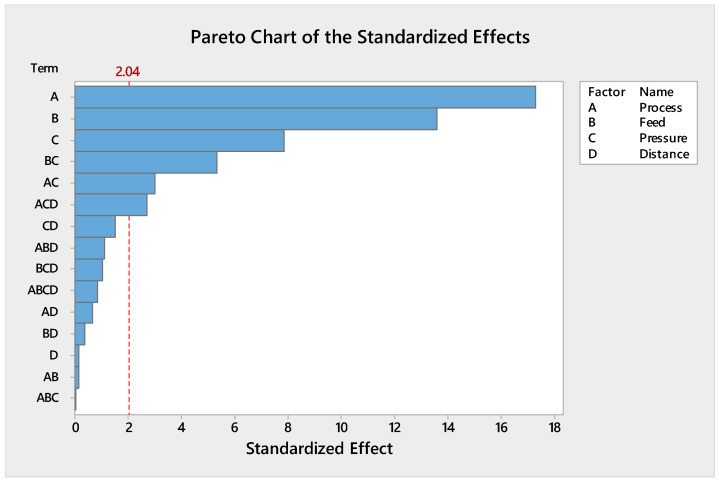
Pareto chart for the adhesion strength.

**Figure 2 polymers-13-01520-f002:**
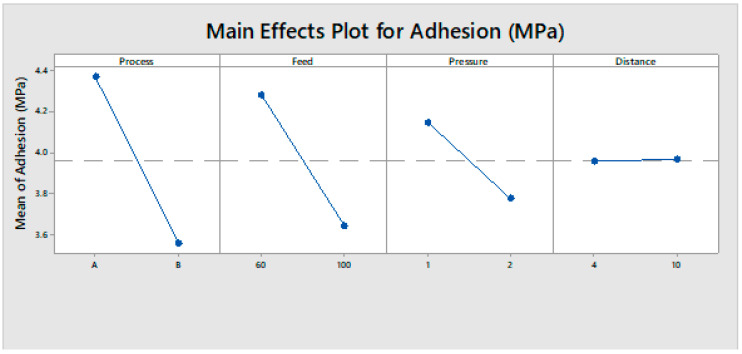
Main effect plot of adhesion strength.

**Figure 3 polymers-13-01520-f003:**
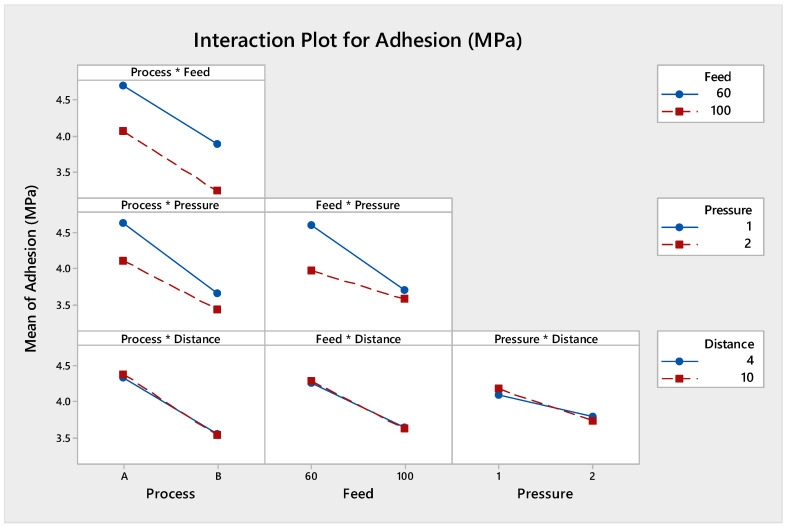
Interaction between process type, treatment feed, pressure, and working distance for adhesion strength.

**Figure 4 polymers-13-01520-f004:**
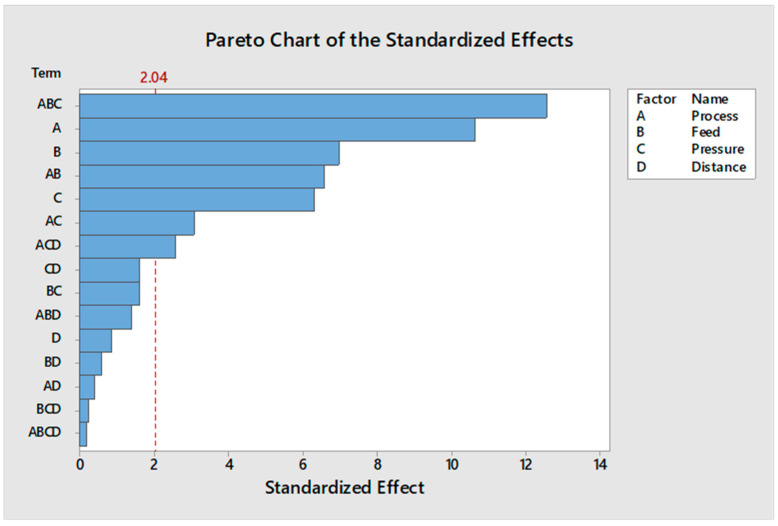
Pareto chart for the effective factors.

**Figure 5 polymers-13-01520-f005:**
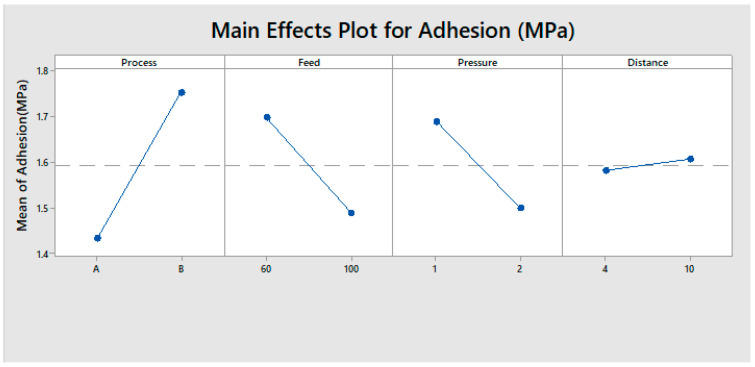
Main effect plot of adhesion strength.

**Figure 6 polymers-13-01520-f006:**
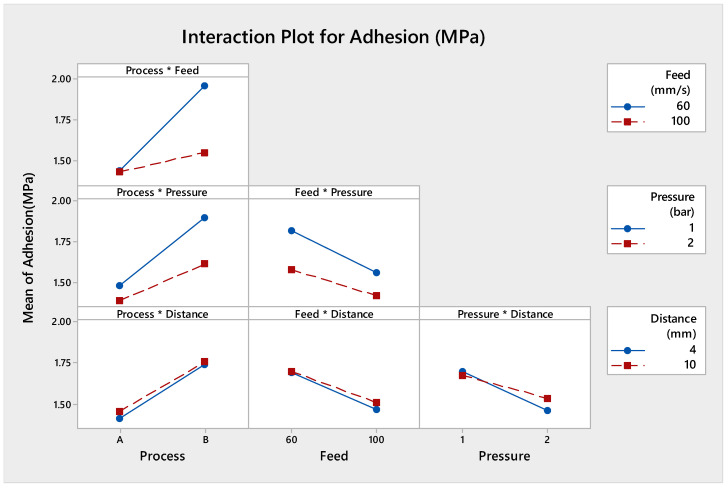
Interaction between process type, treatment feed, pressure, and working distance for adhesion strength.

**Figure 7 polymers-13-01520-f007:**
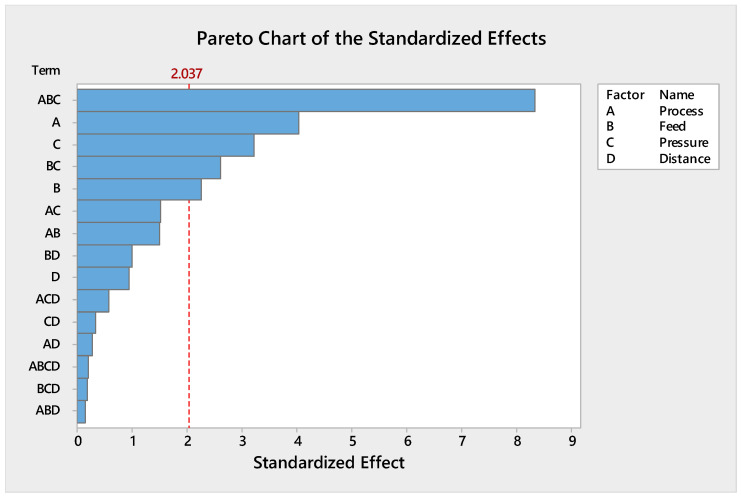
Pareto chart for the colour change.

**Figure 8 polymers-13-01520-f008:**
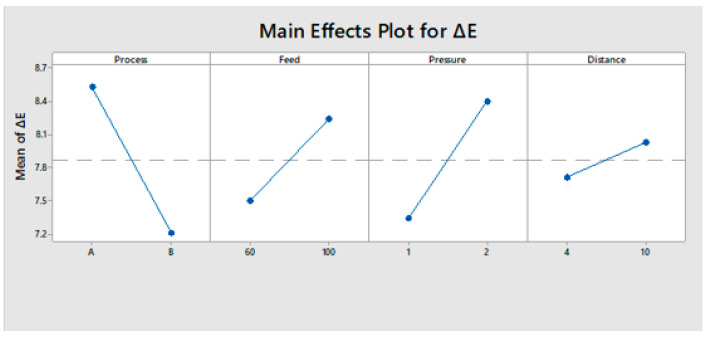
Main effect plot of the colour change.

**Figure 9 polymers-13-01520-f009:**
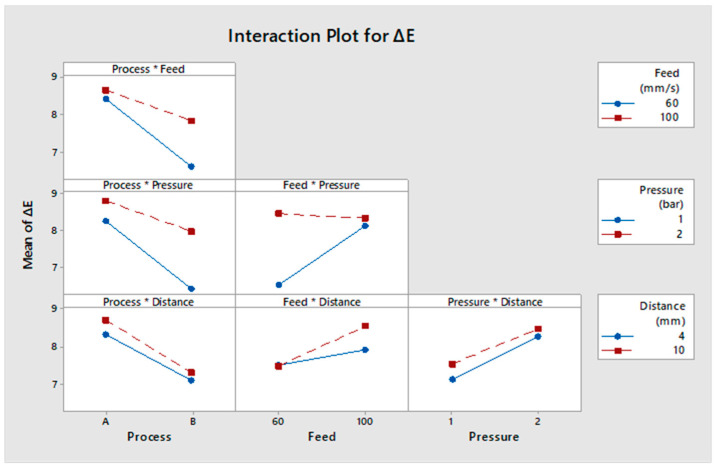
Interaction between process type, treatment feed, pressure, and working distance for colour change.

**Figure 10 polymers-13-01520-f010:**
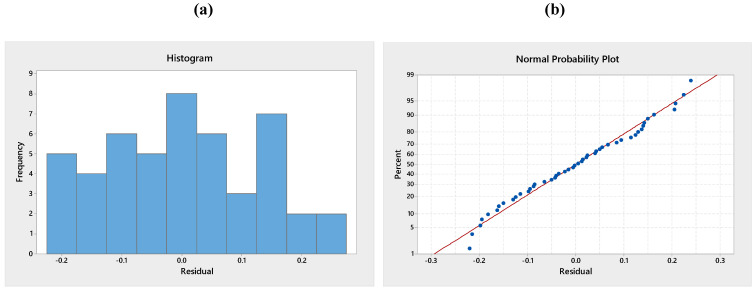
(**a**) Histogram of standardised residuals and (**b**) normal probability plot for adhesion strength before weathering test.

**Figure 11 polymers-13-01520-f011:**
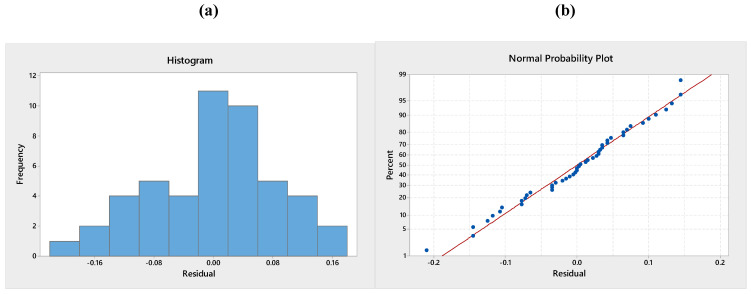
(**a**) Histogram of standardised residuals and (**b**) normal probability plot for adhesion strength after weathering test.

**Figure 12 polymers-13-01520-f012:**
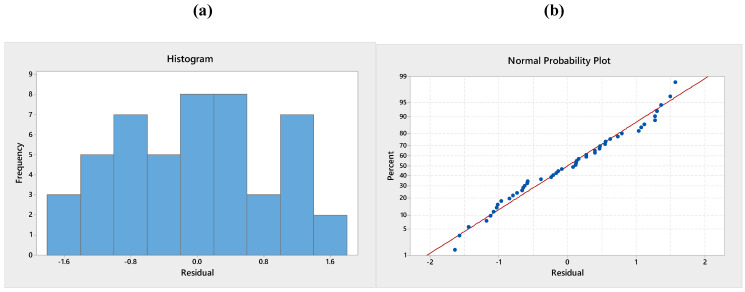
(**a**) Histogram of standardised residuals and (**b**) normal probability plot for colour change.

**Figure 13 polymers-13-01520-f013:**
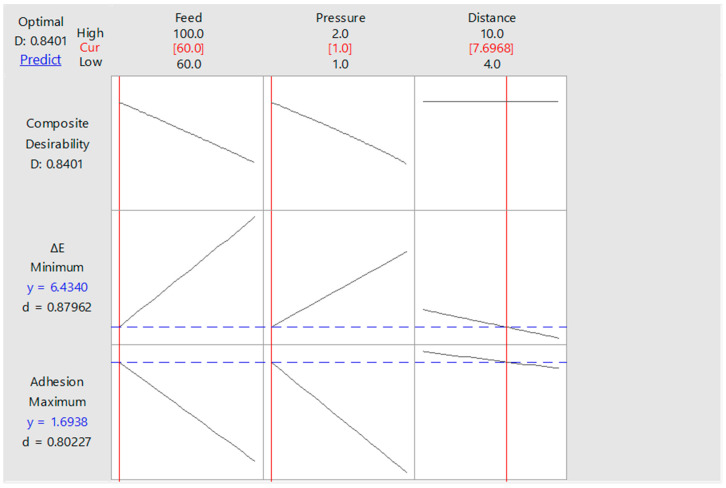
Optimum factor levels for process A.

**Figure 14 polymers-13-01520-f014:**
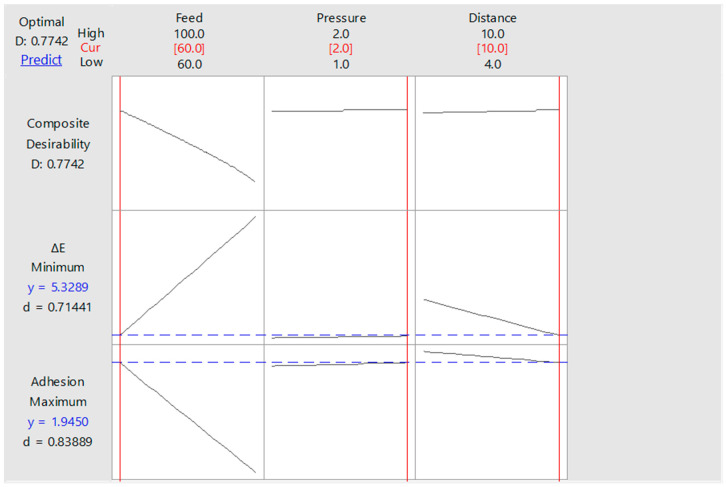
Optimum factor levels for process B.

**Table 1 polymers-13-01520-t001:** Properties of plasma system.

Plasma type	Nozzle type	Generator	Gas	Voltage
Non-Thermal	RD1004	FG3001	Air (114 normal L/min)	500 V

**Table 2 polymers-13-01520-t002:** Feature of plasma system.

Factors	Low-Level	High-Level
Distance	4 mm	10 mm
Feed	60 mm/s	100 mm/s
Pressure	1 bar	2 bar

**Table 3 polymers-13-01520-t003:** Feature of water-based coating.

Layer	Viscosity (20 °C DIN4.sn)	Density (g/cm^3^)	Solid Content (%)
Primer	12	1.02	56
Filling	10	1.08	46
Topcoat	8	1.12	50

**Table 4 polymers-13-01520-t004:** Heat treated and plasma treatment process parameters with levels.

Symbol	Variables	Unit	Level (−1)	Level (+1)
A	Heat treatment process	-	Process A	Process B
B	Treatment feed	mm/s	60	100
C	Pressure	bar	1	2
D	Working distance	mm	4	10

**Table 5 polymers-13-01520-t005:** Results of ANOVA for adhesion strength before artificial weathering test.

Source	*DF*	*Adj SS*	*F*-Value	*p*-Value
Model	15	16.0831	45.82	0.000 *
Linear	4	13.3097	142.20	0.000 *
Process	1	7.0200	300.00	0.000 *
Feed	1	4.3350	185.26	0.000 *
Pressure	1	1.4504	61.98	0.000 *
Distance	1	0.0006	0.03	0.874
Two-Way Interactions	6	0.8937	6.37	0.000 *
Process * Feed	1	0.0005	0.02	0.884
Process * Pressure	1	0.2109	9.01	0.005 *
Process * Distance	1	0.0108	0.46	0.501
Feed * Pressure	1	0.6633	28.35	0.000 *
Feed * Distance	1	0.0030	0.13	0.721
Pressure * Distance	1	0.0532	2.27	0.141
Three-Way Interactions	4	0.2459	2.63	0.053
Process * Feed * Pressure	1	0.0000	0.00	0.979
Process * Feed * Distance	1	0.0280	1.20	0.282
Process * Pressure * Distance	1	0.1700	7.27	0.011 *
Feed * Pressure * Distance	1	0.0253	1.08	0.306
Four-Way Interactions	1	0.0176	0.75	0.392
Process * Feed * Pressure * Distance	1	0.0176	0.75	0.392
Error	32	0.7488		
Total	47	16.8319		

P: error variance, SS: sum of squares, DF: degrees of freedom, and F: F-test value. (*) Term is significant with a 95% reliability interval.

**Table 6 polymers-13-01520-t006:** Results of ANOVA for after artificial weathering test.

Source	*DF*	Adj *SS*	*F*-Value	*p*-Value
Model	15	4.94450	34.29	0.000 *
Linear	4	1.95364	50.81	0.000 *
Process	1	1.08375	112.74	0.000 *
Feed	1	0.46482	48.36	0.000 *
Pressure	1	0.38254	39.80	0.000 *
Distance	1	0.00735	0.76	0.388
Two-Way Interactions	6	0.65587	11.37	0.000 *
Process * Feed	1	0.41344	43.01	0.000 *
Process * Pressure	1	0.09127	9.49	0.004 *
Process * Distance	1	0.00150	0.16	0.695
Feed * Pressure	1	0.02535	2.64	0.114
Feed * Distance	1	0.00304	0.32	0.578
Pressure * Distance	1	0.02535	2.64	0.114
Three-Way Interactions	4	2.02577	52.69	0.000 *
Process * Feed * Pressure	1	1.51504	157.61	0.000 *
Process * Feed * Distance	1	0.01815	1.89	0.179
Process * Pressure * Distance	1	0.06304	6.56	0.015 *
Feed * Pressure * Distance	1	0.00050	0.05	0.820
Four-Way Interactions	1	0.00027	0.03	0.869
Process * Feed * Pressure * Distance	1	0.00027	0.03	0.869
Error	32	0.30760		
Total	47	5.25210		

P: error variance, SS: sum of squares, DF: degrees of freedom, and F: F-test value. (*) Term is significant with a 95% reliability interval.

**Table 7 polymers-13-01520-t007:** Results of ANOVA for colour change.

Source	DF	*Adj* SS	*F*-Value	*p*-Value
Model	15	145.068	8.50	0.000 *
Linear	4	39.828	8.75	0.000 *
Process	1	18.494	16.26	0.000 *
Feed	1	5.780	5.08	0.031 *
Pressure	1	11.854	10.42	0.003 *
Distance	1	1.024	0.90	0.350
Two-Way Interactions	6	15.957	2.34	0.055
Process * Feed	1	2.543	2.24	0.145
Process * Pressure	1	2.648	2.33	0.137
Process * Distance	1	0.089	0.08	0.782
Feed * Pressure	1	7.700	6.77	0.014 *
Feed * Distance	1	1.147	1.01	0.323
Pressure * Distance	1	0.129	0.11	0.738
Three-Way Interactions	4	85.371	18.76	0.000 *
Process * Feed * Pressure	1	79.054	69.51	0.000 *
Process * Feed * Distance	1	0.026	0.02	0.882
Process * Pressure * Distance	1	0.368	0.32	0.574
Feed * Pressure * Distance	1	0.040	0.04	0.852
Four-Way Interactions	1	0.050	0.04	0.835
Process * Feed * Pressure * Distance	1	0.050	0.04	0.835
Error	32	36.396		
Total	47	181.464		

P: error variance, SS: sum of squares, DF: degrees of freedom, and F: F-test value. (*) Term is significant with a 95% reliability interval.

## Data Availability

The data presented in this study are available on request from the corresponding author.

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
