# Peer review of "Improvement of Heat-Treated Wood Coating Performance Using Atmospheric Plasma Treatment and Design of Experiments Method"

_polymers, 2021, doi:10.3390/polym13091520_

Round 1

Reviewer 1 Report

An interesting article on the possibilities of improving the adhesion of watter based varnish to the surface of heat-treated wood. At the same time, the color changes of the used watter based varnish after artificial aging were studied. Changes in these parameters after plasma modification of wood surfaces were compared.

The DOE method to investigate the significant factors and multi objective optimization technique was used to assess the tests.

The article contains links to a considerable number of references. Reference [11] was not cited in the article. Please add or remove from the list of references if it is not relevant.

The solution procedure appears to be correct and described in sufficient detail.

The discussion of the results could have been more extensive, but the description of the results is still acceptable. In view of the above findings, the results of the solution could be used in practice in assessing the possibility of improving the adhesion of the varnish of heat-treated plasma-modified wood surface. Equally interesting, with a practical impact, is the possibility of surface modification with plasma to influence the color changes of treated wood after weathering.

Author Response

Dear Reviewer 1,

Thank you very much for your valuable criticism.
Reference 11 was cited.
The conclusion section was improved.

Reviewer 2 Report

The article analysis the improvement of heat treated wood performance using atmospheric plasma treatment. the article is strong, however, results section is poor in terms of explanation of changes and observations presented. Also, I have few additional remarks:

1) Please read again the whole article and delete the dashes which appear in the middle of some words.

2) There is no information about the supplier and its country of origin of wooden materials and raw materials for the varnish.

3) I do suggest improving discussion section with more detailed explanations and at least percentage variations of the results obtained and explanations of possible reasons for one or another changes. 

4) MPa but not Mpa

5) The whole results section, i.e. mainly the text is missing some results discussion. Because at a current state, it only repeats almost the same sentence "...were statistically significant" or similar. Please expand your observations and support them with numerical values.

Author Response

Thank you very much for your valuable criticism.
Dashes were corrected.
Added information on supplier and country of origin of wood material and varnish raw materials
Discussion section was improved.
The conclusion part was improved.
Mpa's were corrected to MPa.

Reviewer 3 Report

General comment:

The authors of this paper describe the effect of heat treated wood coating performance using atmospheric plasma treatment. I confirm that paper has merit but needs a revision.

Each paragraph is amply described but not correctly discussed. I invite authors to strengthen the discussion section.

Title: It is correctly clear.

Abstract: Abstract requires a minor revision to introduce better the aim of the study and  the comprehension of the work.

Keyword: There are too many words; please reduce it.

Introduction

The Introduction provides a complete state of art what is being discussed in this paper. In this format, the introduction section is complete. Some sentences are obvious and not necessary; other sentences can be useful for discussion section moving from here.

M&M

M&M is complete and correct but I suggest to realize a table with a summary with all parameters monitored.

  1. Results and Discussion is wrong. It is only Results; Discussion is 4 section.

Results reported data obtained but the discussion section was not improved.

Lines 108-110 “Because…here”. Please explain better this sentence.

Section Discussion is not developed; several sentences repeat general concept.

Conclusion

Chapter Conclusion is not correct for this type of journal. The conclusions should be written comparing objects of the study and the results obtained but the authors reported a discussion section The conclusions repeat the results adding considerations useful for discussion paragraph. More future developments and conclusions should be considered.

Author Response

Dear Reviewer,

Thank you very much for your valuable criticism.
Some sentences were added to the introduction section.
Conclusion and discussion section were separated.
Discussion section was improved. The result section was reduced and supported with numerical data.

Keywords were reduced.

Best regards,

Round 2

Reviewer 2 Report

Authors have taken into consideration all my remarks.